# Inflammatory disease progression shapes nanoparticle biomolecular corona-mediated immune activation profiles

Jacob R. Shaw [1], Nicholas Caprio[2], Nhu Truong[2], Mehari Weldemariam[2], Anh Tran [2], Nageswara Pilli[2], Swarnima Pandey[2], Jace W. Jones [2], Maureen A. Kane[2,3] & Ryan M. Pearson [1,2,3] ✉

Polymeric nanoparticles (NPs) are promising tools used for immunomodulation and drug delivery in various disease contexts. The interaction between NP surfaces and plasma-resident biomolecules results in the formation of a biomolecular corona, which varies patient-to-patient and as a function of disease state. This study investigates how the progression of acute systemic inflammatory disease influences NP corona compositions and the corresponding effects on innate immune cell interactions, phenotypes, and cytokine responses. NP coronas alter cell associations in a disease-dependent manner, induce differential co-stimulatory and co-inhibitory molecule expression, and influence cytokine release. Integrated multi-omics analysis of proteomics, lipidomics, metabolomics, and cytokine datasets highlight a set of differentially enriched TLR4 ligands that correlate with dynamic NP corona-mediated immune activation. Pharmacological inhibition and genetic knockout studies validate that NP coronas mediate this response through TLR4/MyD88/NF-κB signaling. Our findings illuminate the personalized nature of corona formation under a dynamic inflammatory condition and its impact on NP-mediated immune activation profiles and inflammation, suggesting that disease progression-related alterations in plasma composition can manifest in the corona to cause unintended toxicity and altered therapeutic efficacy.

Polymeric nanoparticles (NPs) used for immunomodulation have garnered significant attention demonstrating effectiveness in pre-clinical and clinical studies for treating infectious diseases, cancer, autoimmunity, and several others. However, there has been growing concern regarding the rigor and reproducibility of pre-clinical nanoparticle formulations, high interpatient variability in clinical trials, and their overall lack of successful clinical translation[1]. The complexity and dynamics of disease states, as well as the timing of NP administration, pose a formidable challenge to induce desired immune responses. A confounding factor is the interaction between NPs and plasma-resident biomolecules. Upon introduction into biological fluids, biomolecules such as proteins, lipids, and others rapidly adsorb to the NP surface, forming a coating known as the biomolecular corona[2–5]. Corona formation represents a conversion from the "synthetic identity" of the NP to a newly acquired "biological identity," exerting broad effects on the cellular interactions, immune activation, toxicity, and consequently, therapeutic outcomes[5–9]. Therefore, when developing NPs for immunomodulatory applications, it is crucial to

[1]Department of Microbiology and Immunology, University of Maryland School of Medicine, 685 W. Baltimore Street, Baltimore, MD 21201, USA. [2]Department of Pharmaceutical Sciences, University of Maryland School of Pharmacy, 20 N. Pine Street, Baltimore, MD 21201, USA. [3]Marlene and Stewart Greenebaum Comprehensive Cancer Center, University of Maryland School of Medicine, 22 S. Greene Street, Baltimore, MD 21201, USA. ✉e-mail: rpearson@rx.umaryland.edu

understand corona formation, its composition, and how it affects biological activity.

While it is well-established that the corona composition is influenced by the physicochemical properties of the NP, recent observations suggest that host pre-existing conditions can also impact corona composition, termed "personalized biomolecular corona"[10–14]. The disease-specific corona formation is proposed to be a driving factor for the diversity in NP-based therapeutic responses and the high prevalence of infusion reactions observed clinically, occurring in 7% to 20% of patients depending on the formulation and treatment population[15–18]. Characterization of clinical formulations suggest that NPs concentrate disease-related acute inflammatory response proteins when introduced into the blood from patients with systemic inflammatory conditions, like sepsis, which is not observed in healthy controls[19,20]. Additionally, ex vivo protein corona compositions demonstrate high interpatient variability and immune cell uptake within the same disease groups or otherwise healthy controls[11,21]. These differences in disease-specific corona proteins have been utilized for the characterization of disease progression and discovery of biomarkers in several cases; however, the impact that disease progression has on cell recognition and corresponding immune activation profiles induced by NP coronas has not been elucidated[22–26]. The increased prevalence of inflammatory proteins, such as classical opsonins, has direct implications on NP-immune cell recognition, likely leading to an increase in phagocytic clearance and off-target biodistribution and toxicity[27–30].

To bridge this gap in knowledge, here we employed a murine model of severe systemic inflammation to investigate the influence of disease state dynamics on plasma biomolecule compositions (including proteins, lipids, and metabolites) and the corresponding NP corona fingerprints. We examined the associated alterations in innate immune cell interactions, phenotypes, and activation profiles by employing an integrated multi-omics approach to identify and test a series of biomolecules that contributed to the differential immunostimulatory potential of coronas. Considering the high prevalence of heterogeneous inflammatory comorbidities, these findings highlight the need to comprehensively identify disease state-dependent effects of relevant plasma biomolecules, corresponding NP coronas, and their effects on immunomodulation to aid the future development of nanotherapeutic approaches and normalize efficacy among diverse patient populations.

## Results and Discussion

### Inflammation dynamics alter biomolecular corona formation and innate immune cell interactions

Poly(lactic-co-glycolic acid) (PLGA)-based NPs have been investigated for multiple drug delivery and immunomodulatory applications over the past several decades, with several reaching clinical trials and clinical approval[31–36]. In addition, PLGA is recognized by the US Food and Drug Administration (FDA) and the European Medical Association (EMA) as a safe, biodegradable polymer for pharmacological applications[31]. We formulated 200 nm PLGA NPs using a microfluidic hydrodynamic flow-focusing methodology, which enabled scalable and reproducible NP formation via nanoprecipitation[37]. We employed lipopolysaccharide (LPS)-induced endotoxemia as a highly dynamic inflammatory disease state and model of severe systemic inflammation to evaluate NP corona formation and corresponding immunological effects altered by the acquired biological identity, which allowed us to observe distinct immune signatures and enrichment patterns in the corona that may be more subtle under milder conditions but could provide mechanistic insights into NP behavior in inflammatory states. NPs were incubated with pooled plasma collected from LPS-treated mice at various time points and centrifuged to separate unbound plasma biomolecules from NP coronas (Fig. 1a). For representative timepoints, we isolated plasma from Naïve (No-LPS), 3hr-Post LPS, and 8hr-Post LPS mice, hereafter referred to as NaïvePlas, 3hrPlas, and 8hrPlas, respectively. In addition, PLGA NPs were used as a control, which were un-coated and not pre-exposed to plasma. To characterize the effect that inflammation has on plasma dynamics, we conducted a multiplexed cytokine analysis revealing a rapid induction of pro-inflammatory cytokines 3 h post-LPS injection, which gradually diminished over time (Fig. 1b). Aligning with the clinical progression of severe inflammatory diseases, total plasma protein content significantly decreased over time (Fig. 1c). Previous literature has attributed a decrease in total plasma protein to increased vascular permeability, cellular catabolism, decreased protein synthesis, hemodilution, and protein-rich fluid extravasation into tissues[38,39]. After corona formation, distinct NP corona size alterations were observed based on the plasma source, with 8hrPlas NP coronas showing the broadest size distributions and all showing increased zeta potentials compared to the PLGA NP control (Fig. 1d, Supplementary Table 1). The size distributions indicate that individual NP coronas could have distinct fingerprints, which contribute to the formation of larger aggregates. Moreover, Coomassie-stained SDS-PAGE revealed qualitative differences in the protein fingerprints of these coronas, suggesting disease-severity/progression-dependent variations in protein composition (Fig. 1e). Despite the reduction in total plasma protein content (Fig. 1c), the total protein content was unchanged for all coronas likely owing to the saturation of NP surface area (Fig. 1f). The mononuclear phagocyte system (MPS), particularly macrophages, plays a crucial role in recognizing, engulfing, and eliminating NPs, contributing to both the overall clearance of NPs from the body and the development of acute infusion reactions[40]. Using bone marrow-derived macrophages (macrophages), analysis of the association kinetics revealed plasma-dependent alterations in interactions of NP coronas compared to the PLGA NP control (Fig. 1g). Previous studies have also showed alterations in cellular associations due to corona formation, which was explained by a reduction in non-specific interactions due to neutralized zeta potentials and alterations in NP uptake receptors or internalization mechanisms[41–43]. Interestingly, NPs coated with 8hrPlas showed significantly reduced cell association compared to 3hrPlas and NaïvePlas, reaching a maximum of 38% cell positivity after 24 h (Fig. 1g). Previous literature has demonstrated that coronas formed from different individuals' plasma can result in altered cellular uptake kinetics, with some showing a correlation between opsonin adsorption, like immunoglobulins and complement, and cellular association[11,12,44,45]. However, this data suggests that NPs may also have different uptake kinetics within the same individuals depending on the disease context.

### Dynamic NP coronas induce distinct inflammatory profiles

Next, we investigated whether NP coronas derived from dynamic inflammatory states could induce distinct innate immune cell responses by examining alterations in cellular phenotypes and cytokine secretions in macrophages treated with various NP coronas. An elevation of co-stimulatory molecules CD80 (1.43-fold increase) and CD86 (2.30-fold increase), and the co-inhibitory molecule PD-L1 (14.61-fold increase) from 3hrPlas_PLGA-treated macrophages was observed after 24 h, while PLGA and NaïvePlas_PLGA showed no significant differences (Figs. 2a, S1). Interestingly, 8hrPlas_PLGA induced a 5.7-fold increase in PD-L1 presentation, with minimal elevations in CD80 (1.07-fold increase) and CD86 (1.11-fold increase). Moreover, there was an 11-15% increase in cell death for 8hrPlas_PLGA and 3hrPlas_PLGA treated cells, respectively (Supplementary Fig. 1). An elevation of co-stimulatory molecule expression and cell death is often associated with immune stimulation; therefore, we measured the pro-inflammatory cytokine TNFα as a function of time in these supernatants and observed a robust induction in the 3hrPlas_PLGA-treated cell culture supernatants after only 3-h of incubation (Fig. 2b). Further cytokine profiling demonstrated that 3hrPlas_PLGA induced the

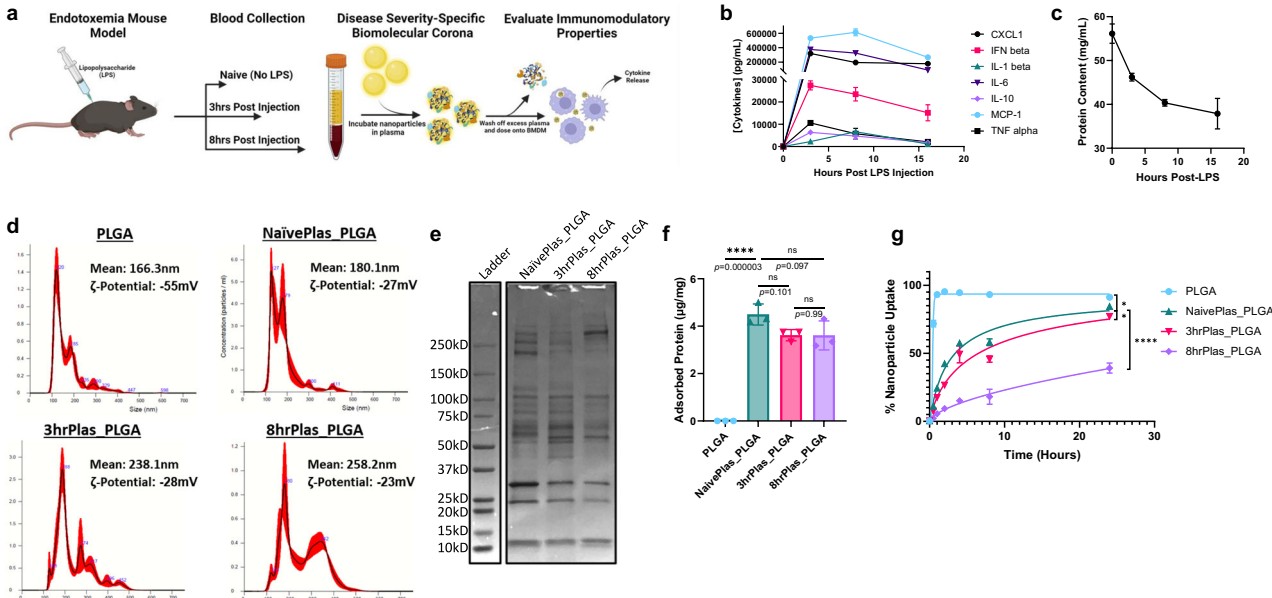

**Fig. 1 | Endotoxemia plasma characterization and biomolecular corona translation. a** Schematic representation of the endotoxemia mouse model and plasma extractions for NP corona formation. Plasmas were extracted after varying time-points to assess different inflammation severities ($n = 10$ mice per timepoint) and used to coat NPs for subsequent macrophage treatments. Created in BioRender. Shaw, J. (2025) https://BioRender.com/s80i128. **b** Pro-inflammatory cytokine profiles of pooled whole mouse plasma after 0, 3, 8, and 16 h-post LPS injection (representing 3 measurements of $n = 10$ pooled mouse plasmas per timepoint). **c** Total protein quantified in whole plasma over time using a bicinchoninic acid (BCA) protein assay ($n = 3$ biological replicates). **d** Nanoparticle Tracking Analysis (NTA) of NP corona diameter size distributions after washing off excess plasma. Plots are representative of $n = 3$ measurements of 30 second recordings. Error bars are represented as red-shaded regions. Mean size and zeta-potential are presented

for each condition. **e** NP corona proteins from NaivePlas, 3hrPlas, and 8hrPlas samples were eluted and run through an SDS-PAGE gel and stained with Coomassie for total protein. **f** Quantification of proteins eluted from NP coronas using a 3-(4-carboxybenzoyl)quinoline-2-carboxaldehyde (CBQCA) protein assay, represented as ng of protein per mg NP ($n = 3$ independent replicates). Significance was calculated using a one-way ANOVA with Tukey post hoc test. **g** Uptake/association kinetic analysis of Cy5.5-labeled PLGA NPs or NP corona-treated macrophages up to 24 h, quantified using flow cytometry ($n = 3$ biological replicates). Significance was determined in the final 24-hour timepoint using a one-way ANOVA with Tukey's post-hoc test (PLGA:NaivePlas_PLGA, $p = 0.0255$; NaivePlas_PLGA:3hrPlas_PLGA, $p = 0.0151$; NaivePlas_PLGA: 8hrPlas_PLGA, $p < 0.00001$). All data sets are presented as mean ± S.D. *$P < 0.05$; **$P < 0.01$; ***$P < 0.001$, ****$P < 0.0001$. ns, not significant ($P > 0.05$). Source data are provided as a Source Data file.

expression of classically pro-inflammatory cytokines TNFα, IL-6, and CXCL1, in addition to IL-10, whereas 8hrPlas_PLGA NP coronas did not elicit a similar response (Fig. 2c). Additionally, there was no significant expression of IL-1β or IFN-β among these NP corona treatments, although the absence of IL-1β could be a result of the lack in NLRP3 inflammasome activation necessary to cleave pro-IL-1β (Supplementary Fig. 2). Given the significant differences in inflammatory profiles induced by the 3hrPlas and 8hrPlas NP coronas, we prepared 16hrPlas_PLGA to evaluate if this cytokine trend continues at a later timepoint. Interestingly, 16hrPlas_PLGA displayed a non-inflammatory profile when evaluating these same cytokines. This led to the hypothesis that an acute inflammatory biomolecule may be elevated at early time points, rendering 3hrPlas_PLGA inflammatory, but not at later time points. Overall, these responses are consistent with the expected phenotype associated with co-stimulatory receptor induction by 3hrPlas_PLGA and the primary co-inhibitory receptor induction by 8hrPlas_PLGA. The inflammatory state-dependent macrophage stimulation highlights the impact of plasma dynamics on corona-mediated cellular interactions and immune activation.

To determine if this phenomenon extended to other NP formulations, we examined the same cytokine responses using poly(lactic acid) (PLA) and PEGylated PLGA (PLGA-PEG) NPs. Poly(ethylene glycol) (PEG) is a commonly used steric coating known to deter corona formation, aiding in NP stability, and reducing immune recognition[11]. As expected, PEGylation limited corona-induced alterations in size distributions, compared to their non-PEGylated counterparts (Supplementary Fig. 3, Supplementary Table 1). In addition, SDS-PAGE analysis of corresponding corona fingerprints showed a marked reduction in adsorbed proteins (Supplementary Fig. 4). However, despite the

different polymer core or PEGylation, plasma-specific trends of pro-inflammatory cytokines persisted, with the exception of 16hrPlas_PLGA-PEG (Fig. 2c). 16hrPlas_PLA and 16hrPlas_PLGA-PEG resulted in a slight elevation of cytokines in a comparable pattern to 3hrPlas_PLGA coronas. Correlation matrix and principal component analysis (PCA) of these cytokine fingerprints revealed clustering of 3hrPlas and 16hrPlas NP coronas, independent of NP composition, while PLGA, NaïvePlas, and 8hrPlas NP coronas clustered similarly (Fig. 2d, Supplementary Fig. 2a–d). To determine if the surfactant used during the formulation of these NPs contributed to the immune activation of NP coronas, we also formulated similarly sized PLGA NPs with poly(vinyl alcohol) (PVA) which is a clinically used neutral surfactant. Formation of disease-specific coronas with PLGA-PVA NPs showed a similar dynamic TNFα induction profile in macrophages as the other NP coronas evaluated (Supplementary Fig. 2e). PLGA/PLA NPs have been extensively used as drug delivery vehicles for the treatment of inflammatory diseases, including pre-clinical models of endotoxemia, allergy, experimental autoimmune encephalomyelitis, and several others with varying levels of success[46–48]. Our group and others have shown that similar NPs possess inherent anti-inflammatory and immunomodulatory properties; however, the timing of NP administration was found to contribute to their ability to mitigate lethality in endotoxemia[49–51]. This observation could, in part, be related to the formation of an immunomodulatory corona, which could impede their ability to reduce inflammation by altering cellular uptake and immune activation. Future studies may benefit from evaluating the influence of NP formulation and purification parameters, PEG-alternative coatings, and other strategies to control the potential formation of immunostimulatory NP coronas and improve NP efficacy.

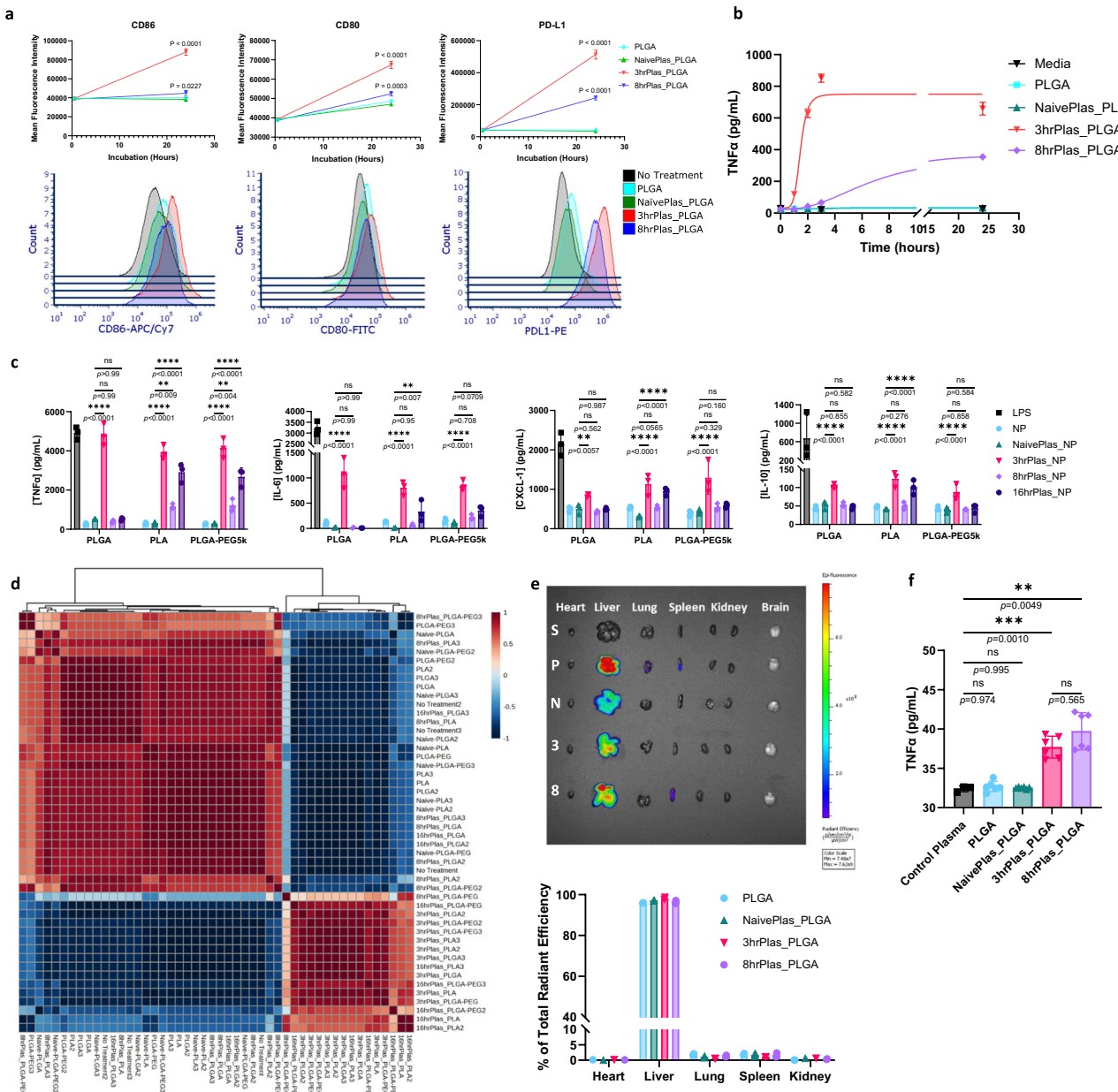

**Fig. 2 | Dynamic immune recognition of NP corona complexes. a** Flow cytometry mean fluorescence intensity (MFI) quantification of macrophage surface markers CD86, CD80, and PD-L1 after treatment with pristine PLGA NPs or NP coronas for 30-min or 24-h (n = 3 biological replicates per timepoint). Representative histograms of 24-hour timepoints are shown below. Data is presented as mean values +/- SD. Statistical significance was determined with a two-way ANOVA using a Dunnett's test with No Treatment set as control. **b** TNFα secretions over time from macrophages after PLGA NP or NP corona treatment (n = 3 biological replicates per timepoint). Data is presented as mean values +/- SD. **c** Multiplex cytokine analysis in the supernatants of NP corona-treated macrophages after 3-hour incubation (n = 3 biological replicates). Data is presented as mean values +/- SD. Statistical significance was determined with a two-way ANOVA using Tukey post hoc test. **d** Spearman correlation matrix of the macrophage multiplex cytokine analysis.

Colors range from dark red, indicating a strong positive correlation between sample cytokines, to dark blue, indicating strong negative correlation. **e** Organ biodistribution of Cy5.5-labeled PLGA NPs coated in various plasma coronas 3-h post i.v. injection in naïve C57BL/6 mice. Heart, liver, lung, spleen, kidney, and brain were isolated and analyzed via in vivo imaging system (IVIS) for NP fluorescence. Representative organ samples are presented from saline (S), PLGA (P), Naïve-Plas_PLGA (N), 3hrPlas_PLGA (3), and 8hrPlas_PLGA (8) treated mice. Percent of total radiant efficiency is presented in the graph below (n = 2 mice per group). No fluorescence was detected in brain samples. **f** TNFα plasma concentration from NP corona-treated IVIS mice (n = 2 mice per group, 3 technical replicates each). Data is presented as mean +/- SD. Significance was calculated using a two-way ANOVA with Tukey post hoc test. *P < 0.05; **P < 0.01; ***P < 0.001, ****P < 0.0001. ns, not significant (P > 0.05). Source data are provided as a Source Data file.

Alterations in NP size distributions, uptake kinetics, and immune cell recognition are often associated with differential biodistribution in vivo[29,52]. To confirm if the pro-inflammatory phenotypes observed ex vivo held in vivo, fluorescently-labeled PLGA NPs were prepared with various coronas and intravenously injected into naïve C57BL/6 mice. There were no observable differences measured in organ

distribution between various NP coronas, with NPs primarily localizing in the liver (Fig. 2e). The NP sizes used in our studies, ranging from 160–260 nm, is likely the primary factor influencing liver uptake, however, it is possible that the different coronas could alter specific cellular interactions within organs as shown in other studies[52–54]. Interestingly, the 3hrPlas and 8hrPlas derived NP coronas significantly

increased plasma TNFα levels (Fig. 2f). The difference for 8hrPlas compared to the in vitro findings can be attributed to 8hrPlas NP coronas likely stimulating other immune cells in vivo. Taken together, these findings demonstrate the differential immunostimulatory capabilities of various NP coronas caused by NP composition and disease state dynamics, suggesting that the differential presence of biomolecules in coronas are contributing to the measured alterations in immunological profiles.

Lastly, it is well-known that coronas are formed rapidly upon plasma introduction and dynamically change over time and in our current studies NP coronas were formed by incubating NPs in plasma for 1 hour[55,56]. Therefore, we assessed the inflammatory potential of 3hrPlas_PLGA by altering NP plasma incubation times. After only 3.72 min of plasma incubation, 3hrPlas NP coronas were able to induce half-maximal TNFα response in macrophages compared to the 1-hour control (Supplementary Fig. 5). Given this, we asserted that the responsible pro-inflammatory biomolecule(s) have a high affinity for PLGA NPs and rapidly reach saturation after being introduced into the plasma.

## Multi-omic analysis for biomolecule identification

We performed comprehensive proteomic, lipidomic, and metabolomic mass spectrometry analyses to identify the biomolecules responsible for the differential corona-mediated immune responses. We noted a time-dependent decrease in plasma triacylglycerols, lysophosphatidylcholines, carbohydrates, and amino acids, accompanied by an increase in phosphatidylcholines, organic acids, phospholipids, and uncategorized "Other" proteins, primarily composed of cell-free hemoglobin (Fig. 3a-c). Dynamics in the percentage of plasma protein categories are often challenging to discern due to the substantial presence of albumin, which constitutes approximately 60% of the total protein content and tends to overshadow other protein alterations. These differences became more apparent when examining the coronas, as they exhibited lower albumin abundance, potentially influenced by the negative charge of albumin being repelled by the negatively charged NPs. When comparing corona fingerprints, significant elevations in clotting factors, inflammatory response proteins, cytoskeletal proteins, phosphatidylcholines, sphingomyelins, lysophosphatidylcholines, and fatty acids were observed over the course of disease progression (Fig. 3d-f). The elevation of clotting factors and cytoskeletal proteins may explain the increase in larger NP aggregate populations observed in the 3hrPlas and 8hrPlas derived coronas, which other groups have similarly observed[57,58]. Importantly, an elevation of corona-bound lysophosphatidylcholines, clotting factors, inflammatory response proteins, and cytoskeletal proteins can be associated with the induction of inflammatory signaling[59–61].

Further analysis of the proteomics dataset revealed a total of 1021 identifiable proteins. Of these, 640 were quantifiable in plasma, while 945 were present in NP coronas, a 47.6% increase (Supplementary Fig. 6a,b). Over 300 proteins were identified in the NP coronas that were not quantifiable in whole plasma variants. In addition, plotting protein plasma abundance against corona-plasma fold-change confirmed that proteins with low plasma abundance were enriched in the NP coronas in alignment with previous literature (Supplementary Fig. 6c-e)[24,62]. Comparing unique proteins observed in each sample, there were 830 proteins shared between all NP coronas and 14 proteins specific to the 3hrPlas samples, which may be responsible for the observed inflammatory cytokine induction. Beyond proteins, we used volcano plots to present the differential abundances across all biomolecules quantified in the coronas, consisting of 830 similar proteins combined with the lipidomics and metabolomics datasets. When comparing Log2 fold changes between 3hrPlas_PLGA and NaïvePlas_PLGA NP coronas, a significant elevation of acute-phase response proteins such as IL-1R antagonist protein, HSP70, Serpin B6, PTX3, and lactoferrin were measured in 3hrPlas_PLGA samples (Fig. 3g).

Additionally, when comparing 3 hrPlas_PLGA to 8hrPlas_PLGA coronas, a marked elevation of classical opsonins such as immunoglobulins and complement proteins were identified in 3hrPlas_PLGA, which may facilitate the cytokine induction (Fig. 3h). Finally, when comparing 8hrPlas_PLGA with NaïvePlas_PLGA NP coronas to validate the overall reduction of opsonins on the 8hrPlas_PLGA samples (Fig. 3i), a marked reduction of complement and immunoglobulins was observed. Further evaluation of individual complement factors highlighted the significant reduction specific to 8hrPlas coronas (Supplementary Fig. 7). Because phagocytic cells, like macrophages, utilize FC-receptor-mediated uptake of opsonized particles, the significant reduction in 8hrPlas_PLGA uptake observed earlier may be explained by a decrease in opsonin binding. Prior literature suggests that during severe inflammation, complement levels can decrease due to their overuse in immune complex formation and subsequent depletion[63–65]. This depletion in serum complement can subsequently alter NP corona complement, and cellular uptake[21,66]. Therefore, although the host plasma is laden with pro-inflammatory molecules at the 8hrPlas timepoint, and NPs still bind inflammatory proteins, their lack of cellular association could reduce the induction of pro-inflammatory cellular responses when compared to 3hrPlas_PLGA samples. In conjunction, these results show that there is likely a compilation of biomolecules either specific to, or significantly elevated in the 3hrPlas NP coronas that have the propensity to induce inflammatory cytokines, and while 8hrPlas NP coronas also contain similar pro-inflammatory biomolecules, the marked reduction in opsonins may reduce their potency.

Given our comprehensive multi-omics datasets and unknown driver biomolecules, we utilized Ingenuity pathway analysis (IPA) to integrate proteomic, lipidomic, metabolomic, and cytokine datasets with the goal to predict biomolecules in the 3hrPlas coronas that can explain the macrophage cytokine fingerprints (Fig. 3j). Using TNFα, IL-6, CXCL1 and IL-10 as downstream outcomes, we generated an upstream biomarker prediction array to visualize potential nodes of interest, where Toll-like receptor (TLR) signaling and immune complex activation by Fc-receptors were identified as contributors to the macrophage cytokine responses observed (Fig. 3k).

## Immunostimulatory coronas induce cytokine release through TLR4 activation

To validate the biomolecule predictions from IPA, we first aimed to identify the signaling pathway, starting with identifying driver transcription factors. NF-κB activation was assessed using a RAW 264.7 macrophage reporter cell line. A significant increase in NF-κB activity with 3hrPlas_PLGA treated cells was observed, but not to the same extent as the LPS control (Fig. 4a). No other NP coronas induced significant NF-κB activity. To recapitulate these results in primary macrophages, Bay11-7082 (Bay11), a small molecule inhibitor of NF-κB, was used to prevent cytokine expression[67]. Bay11 treatment completely prevented TNFα secretions from 3hrPlas_PLGA treated macrophages, validating that the cytokine induction is regulated through NF-κB (Fig. 4b).

Given that numerous damage-associated molecular patterns (DAMPs) were identified in the 3hrPlas_PLGA corona to activate TLRs, and working upstream from NF-κB, we evaluated MyD88-mediated TLR signaling. We used a pharmacological inhibitor, TJ-2010-5, to prevent the homodimerization of MyD88 necessary for signal transduction[68]. Pretreatment of macrophages with TJ-2010-5 resulted in complete prevention of 3hrPlas_PLGA-induced TNFα secretions, in addition to a significant reduction in the LPS control (Fig. 4c). TNFα secretions in the LPS control is to be expected, since LPS signaling is known to proceed through MyD88-independent activation[69]. The complete prevention of TNFα secretions following 3hrPlas_PLGA suggested that the pro-inflammatory responses induced by 3hrPlas_PLGA were MyD88-dependent. Next, macrophages were isolated from TLR4 knockout (K/O) mice to test if the NP corona is activating the pro-

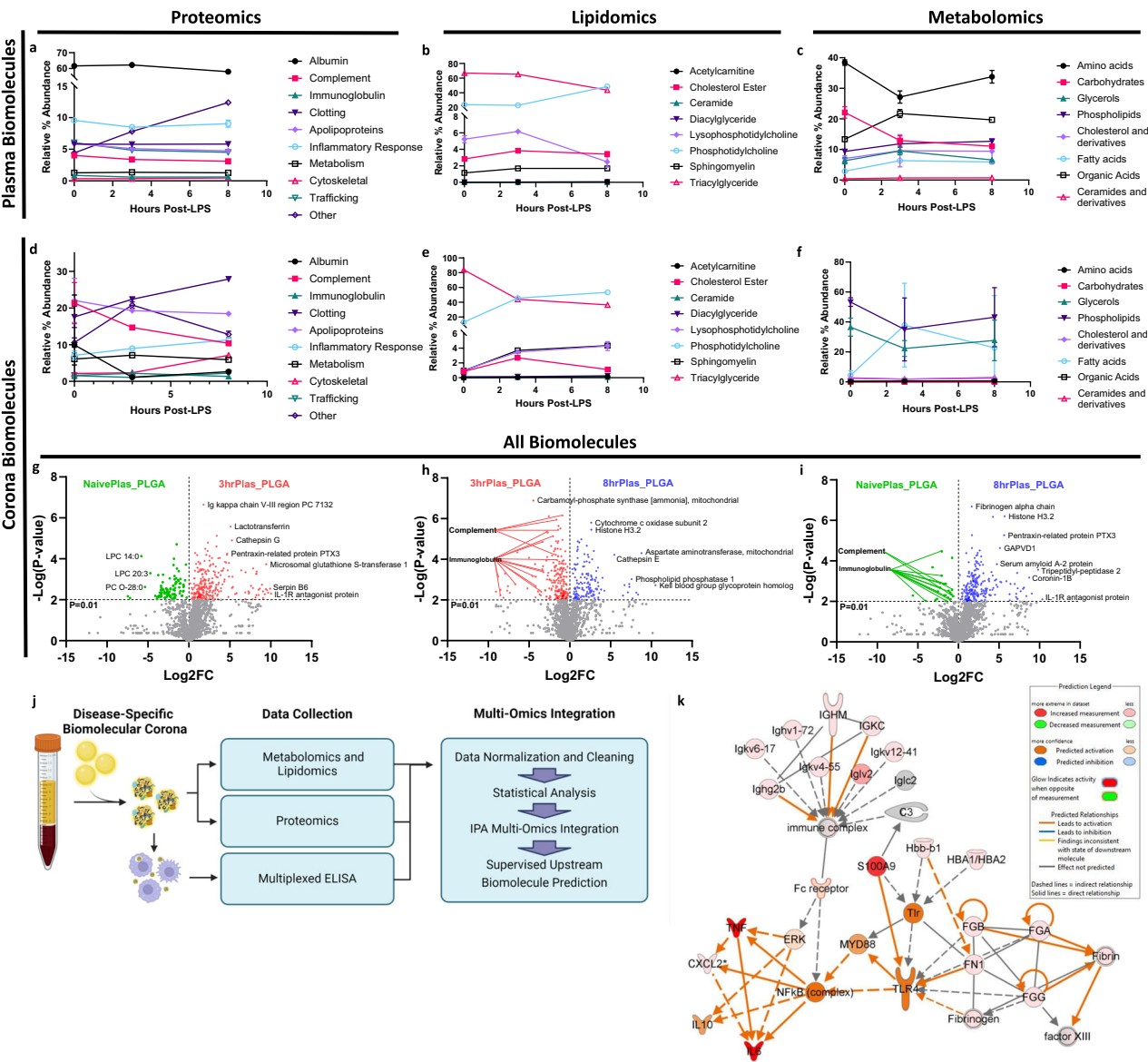

**Fig. 3 | Inflammation severity-associated biomolecule dynamics and corona-induced inflammation prediction.** Bioinformatic quantification of mouse plasma proteins **a**, lipids **b**, and metabolites **c** at varying times after LPS injection ($n = 3$ measurements of pooled plasma from 10 mice/condition/replicate). Bioinformatic quantification of PLGA NP corona proteins **d**, lipids **e**, and metabolites **f** after plasma incubation ($n = 3$ biological replicates). Biomolecules are grouped according to function process or structural category and represented as a percent of total biomolecule abundance. All mass spectrometry data is presented as mean values +/- SD. Volcano plot comparing multi-omic biomolecule abundances between NaïvePlas vs. 3hrPlas_PLGA NP coronas **g**, 3hrPlas vs. 8hrPlas_PLGA NP coronas

**h**, and 8hrPlas vs. NaïvePlas_PLGA NP coronas **i** Volcano data points are colored green (NaïvePlas), red (3hrPlas), and blue (8hrPlas) based on their elevated plasma condition. Significance was determined for each omics dataset individually using a Two-way ANOVA with Benjamini−Hochberg adjustment. **j** Schematic representation of multi-omic data processing and Ingenuity Pathway Analysis (IPA) application. Created in BioRender. Shaw, J. (2025) https://BioRender.com/g93r882. **k** Upstream network effects that were supervised to predict 3hrPlas corona-induced cytokines. Color key and symbols are reported in the legend. Source data are provided as a Source Data file.

inflammatory response through this receptor. The use of TLR4 K/O macrophages resulted in a significant reduction of TNFα in the LPS control and a complete reduction of TNFα secretions in 3hrPlas_PLGA treated cells (Fig. 4d). It is well known that LPS is a potent TLR4 activator, but impure LPS can also signal through TLR2, likely explaining the low TNFα expression in the LPS control[70]. The complete reduction of TNFα following NP corona treatments provided strong evidence that the 3hrPlas_PLGA corona-specific biomolecule(s) target TLR4 (Fig. 4e).

Given the potential that exogenously administered LPS, a potent TLR4 agonist, from the disease model could contribute to the pro-inflammatory NP corona response observed, we quantified the

abundance of LPS in the plasma and coronas of these samples. We observed a significant abundance of LPS in mouse plasma, peaking at 3 h, which aligns with previous literature findings (Supplementary Fig. 8a)[71]. However, when quantifying the LPS bound to NP coronas, we found that 8hrPlas_PLGA had a higher LPS abundance (5.7 EU/mg) compared to 3hrPlas_PLGA (2.6 EU/mg) (Supplementary Fig. 8b). Additionally, when we performed a binding analysis using FITC-labeled LPS, we detected significantly more LPS binding in the 8hrPlas coronas compared to the 3hrPlas coronas, consistent with our earlier endotoxin quantification results (Supplementary Fig. 8c). This elevation in LPS levels on 8hrPlas_PLGA seemed contradictory to the inflammatory response observed in 3hrPlas_PLGA and the absence of such a

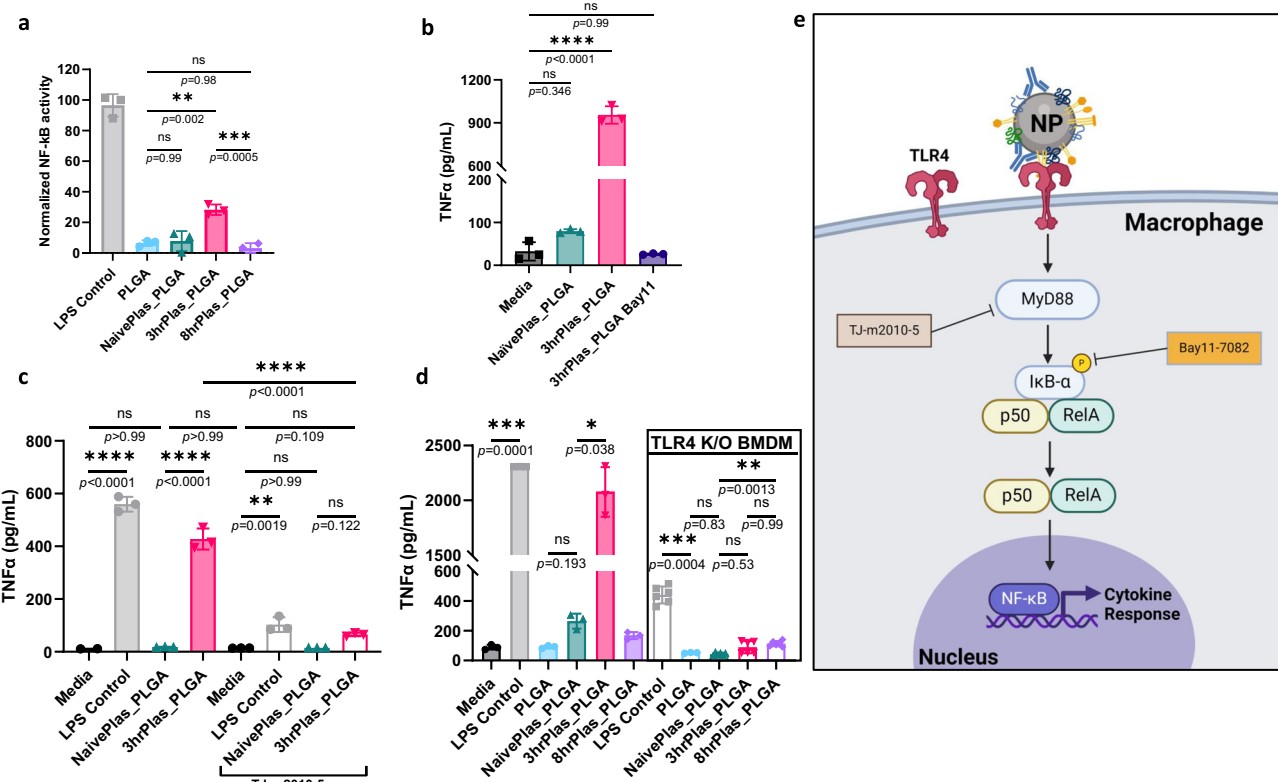

**Fig. 4 | 3hrPlas coronas signal pro-inflammatory cytokine secretions through TLR4 activation of NF-κB. a**. NF-κB activation in RAW-Blue reporter cell line upon incubation with PLGA NP coronas. NF-κB activation was normalized to the LPS positive control (*n* = 3 biologically independent experiments). **b** TNFα secretions from macrophages treated with PLGA NP coronas in the presence or absence of Bay11–7082, an NF-κB inhibitor at 10 μM (*n* = 3 biological replicates), *p* < 0.0001. **c**. TNFα secretions from macrophages treated with PLGA NP coronas in the presence or absence of TJ-m2010-5, a MyD88 inhibitor at 30 μM (*n* = 3 biological replicates).

**d**. TNFα secretions from macrophages (*n* = 3 biological replicates) compared to TLR4 knock-out (K/O) macrophages (*n* = 6 biological replicates) treated with PLGA NP coronas. **e**. Schematic illustration of the proposed corona-dependent pro-inflammatory cytokine signaling. Created in BioRender. Shaw, J. (2025) https:// BioRender.com/v63l645. Data sets are presented as mean ± S.D. Significance for all graphs were calculated using a one-way ANOVA with Tukey's post-hoc test. *P < 0.05; **P < 0.01; ***P < 0.001, ****P < 0.0001. ns, not significant (P > 0.05). Source data are provided as a Source Data file.

response in the former. Based on the EU quantification, the amount of LPS bound to 3hrPlas or 8hrPlas NP coronas was 39 pg/mL and 87 pg/ mL, respectively. To determine if this LPS concentration could induce a pro-inflammatory response, we introduced free LPS at this concentration to macrophages and quantified TNFα secretions compared to NP corona controls. Importantly, there was no significant induction of TNFα at the LPS concentration used (Supplementary Fig. 9a). We further confirmed this lack of response using an NF-κB reporter cell line, where a vast excess of LPS (1 μg/mL) was spiked into NaïvePlas, incubated with PLGA NPs, and the resulting NP coronas were separated from unbound biomolecules and introduced into cells for NF-κB activation measurements. There was no significant induction of NF-κB compared to the proinflammatory 3hrPlas_PLGA control (Supplementary Fig. 9b). It is further unlikely that LPS is the driving biomolecule because NP coronas did not induce IFN-β like the LPS control (Supplementary Fig. 2b). Although it is possible that LPS may contribute to the inflammatory response, our data supported that it is unlikely to be the driving biomolecule in the 3hrPlas NP coronas.

## Hemoglobin and fibrinogen contribute to NP corona-mediated immune responses

Having validated that corona-induced cytokine secretions are mediated through TLR4 signaling, we revisited the multi-omics analysis. We filtered the 3hrPlas-elevated corona-associated biomolecules based on their predicted activation of TLR4 and cytokine profile induction (Fig. 5a). In total, 21 biomolecules were predicted to activate TLR4

(Table 1). Among them, hemoglobin (Hb), fibrinogen (Fb), and fibronectin were the most abundant in the coronas, ranking within the top 10 of all identified proteins (Supplementary Table 2). However, their fold-changes compared to non-inflammatory NaivePlas_PLGA coronas were modest (1.6 to 2-fold increase). S100A9 was strongly predicted to be a causal agent, given its absence in NaïvePlas coronas, its ability to bind to TLR4, and its similar cytokine induction profile. Consequently, we selected Hb, Fib, and S100A9 for further testing. This was achieved by pre-coating PLGA NPs with the recombinant proteins of interest and culturing with macrophages to assess TNFα as a representative cytokine (Fig. 5b). The exception to this approach was S100A9, which we tested by adding an S100A9 inhibitor (Paquinimod) to the 3hrPlasma before NP corona formation, inhibiting the ability of free-floating S100A9 to bind TLR4[72,73]. Hb itself increased TNFα secretions; however, no significant elevation was observed for Hb NP coronas (Hb_High_PLGA) and only a minor elevation was observed when high concentrations of Hb were spiked into Naïve plasma for NP corona formation (Naïve-Hb_PLGA) (Fig. 5c). This suggests that while cell-free hemoglobin is known to be pro-inflammatory[74], hemoglobin sequestration in NP coronas may shield its ability to bind to cell receptors, preventing cytokine induction. On the other hand, Fb pre-coatings significantly elevated TNFα secretions in a dose-dependent manner, while Naïve plasma spiking showed significant, yet reduced, TNFα induction. Fb binding to NP coronas and unfolding has been linked to the induction of pro-inflammatory cytokines, however the magnitude observed was significantly less than 3hrPlas_PLGA[8]. Lastly, the use of

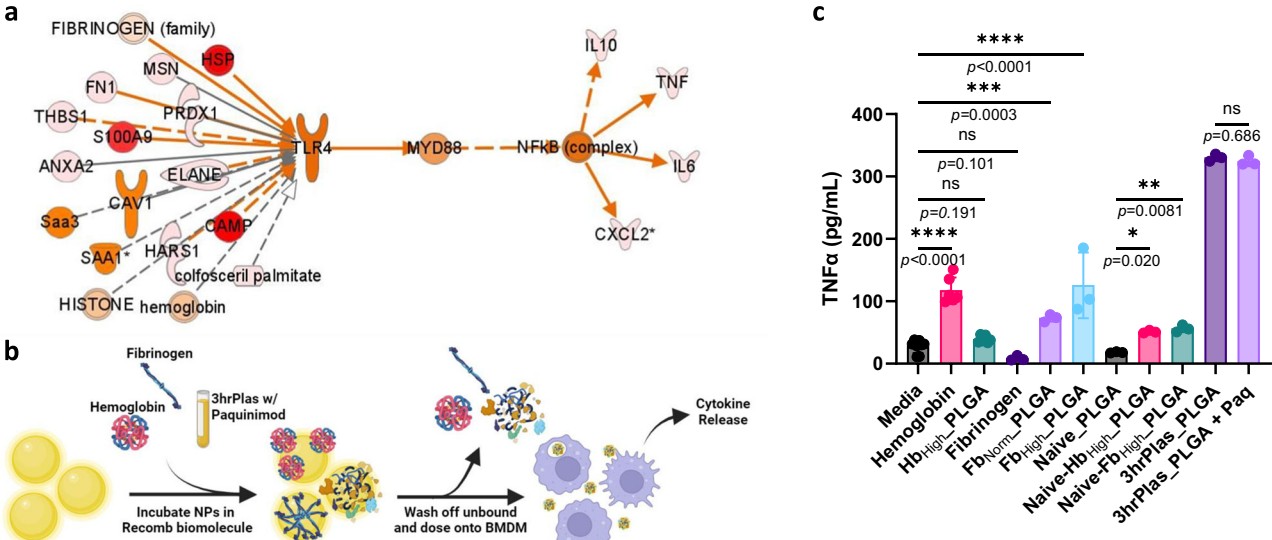

**Fig. 5 | TLR4 ligand NP coatings contribute to pro-inflammatory cytokine induction. a** Upstream network prediction further restricted to 3hrPlas-specific corona biomolecules that induce TLR4/MyD88/NF-kB activation. **b** Schematic representation of recombinant protein pre-coatings to evaluate potential pro-inflammatory biomolecules. Created in BioRender. Shaw, J. (2025) https://BioRender.com/u64r124. **c** Pro-inflammatory TNFα secretion of macrophages treated with various pre-coated NP coronas. NP coronas evaluated include hemoglobin (20 mg/mL) coatings (Hb_High_PLGA), fibrinogen normal (2 mg/mL) and high (10 mg/mL) (Fb_Norm_PLGA, Fb_High_PLGA), 3hrPlas_PLGA spiked with Paquinimod S100A9 inhibitor, along with healthy (Naïve) plasma coatings spiked with hemoglobin or fibrinogen. Data sets are presented as mean ± S.D. (Measurements are from $n = 3$ biological replicates for all samples, with the exception of Media, Hemoglobin, and Hb_High_PLGA which are $n = 6$). Significance was calculated using a one-way ANOVA with Tukey's post-hoc test. $*P < 0.05$; $**P < 0.01$; $***P < 0.001$, $****P < 0.0001$. ns, not significant ($P > 0.05$). Source data are provided as a Source Data file.

Paquinimod to inhibit S100A9 resulted in no significant TNFα differences compared to the 3hrPlas_PLGA control, although, we did not test whole S100A9 protein coatings, which may yield a different response. Taken together, while Hb and Fb may contribute to corona-mediated inflammation, there is likely an additive effect between multiple pro-inflammatory biomolecules leading to the immune activation observed.

To conclude, our work sought to characterize how the progression of systemic inflammation can alter plasma and NP corona compositions, and subsequent immune interactions and activation profiles. Despite efforts to mitigate corona formation, plasma biomolecules readily adsorb to the NP surfaces, which can result unexpected biological responses and efficacy. In this study, we used an endotoxemia mouse model, which represents a severe case of systemic inflammation to demonstrate the impact of inflammation dynamics on NP corona-mediated biological activity. We found that depending on the timepoint for plasma isolation (i.e. measure of disease progression), NP coronas either induced pro-inflammatory phenotypes in macrophages or evaded immune recognition. The dynamic pro-inflammatory responses induced by NP coronas were confirmed in experiments employing three polymer-based NP formulations (PLGA, PLGA-PEG, and PLA) and anionic (PEMA) and neutral (PVA) surfactants. Comprehensive proteomic, lipidomic, and metabolic analyses with multi-omic integration identified a set of acute-phase response biomolecules in the 3hrPlas NP coronas that were associated with inflammatory responses and that corona-induced inflammation signaled through TLR4, MyD88, and NF-κB. These results indicate that elevations in inflammatory plasma biomolecule concentrations translated to NP coronas, in turn imparting an acquired immunostimulatory identity, which can dynamically change over the course of the disease. Hb and Fb were found to partly contribute to the immune activation by NP coronas, suggesting that their presence in the plasma should be assessed when considering specific NP immune activation testing prior to in vivo administration. However, there are some limitations to this work, which future investigations should consider and address. First, the NP corona-induced immune activation phenomenon should be validated using additional NP types as our study primarily focused on polymer-based NPs since different formulations have been shown to adsorb distinct compositions of biomolecules and would be predicted to induce differential biological responses[24,75,76]. Second, although there is previous literature demonstrating disease-specific protein corona composition differences in human plasmas, evaluating the effects of disease dynamics on corona-dependent immune recognition in additional disease models could help to fine-tune the administration of NPs by identifying an optimal therapeutic window where coronas do not induce immune activation or possibly limit therapeutic efficacy. Third, it is likely that many patients will be medicated with other therapeutic agents that could affect the composition of the NP corona and corresponding responses. Overall, this work demonstrates that NP coronas are formed differentially due to disease progression, which significantly affects corona formation, cellular interactions, and corresponding immune activation profiles. These findings contribute to the nanomedicine field by providing evidence that disease-relevant NP coronas should be assessed on a formulation-specific basis to enhance preclinical testing and potential clinical translation. Consequently, previously unsuccessful NP formulations may benefit from re-evaluation with more precise classifications of patient subpopulations and disease-relevant corona characterization.

## Methods

### Materials, cell lines, and animals

Acid-terminated 50:50 PLGA (~0.17 dL/g inherent viscosity in hexafluoro-2-propanol; approximately MW 4.2 kDa) and acid-terminated poly(DL-lactide) (PLA) (approximately MW 11.3 kDa) with low inherent viscosity between 0.16–0.25 dL/g (Product No. B6014-1) were purchased from Lactel Absorbable Polymers (Birmingham, AL). Poly(-ethylene-alt-maleic anhydride) E400 (PEMA, MW 400 kDa, Part No. 105564) was received as a gift from Vertellus™ (Indianapolis, IN).

**Table 1 | Predicted pro-inflammatory biomolecules in the 3hrPlas_PLGA Coronas**

| Biomolecule | Abbreviation | NaïvePlas_PLGA | 3hrPlas_PLGA | 3hr/Naïve Fold Change |
|---|---|---|---|---|
| Hemoglobin alpha/beta[#] | Hb a/b | 7368037972 | 1.6183E+10 | 2.19 |
| Fibrinogen alpha/beta/gamma[#] | Fb a/b/g | 3240853846 | 5137491963 | 1.58 |
| Fibronectin | FN1 | 3038841781 | 5094975764 | 1.67 |
| Serum amyloid A-1 protein | Saa1 | 80273360 | 909709588 | 11.33 |
| Thrombospodin-1 | THBS1 | 232968616 | 902048653 | 3.87 |
| Cathelicidin antimicrobial peptide | CAMP | 9455725 | 390457198 | 41.29 |
| Heat shock cognate 71 kDa protein | HSPA8 | 47700503 | 88425701 | 1.85 |
| Serum amyloid A-3 protein | Saa3 | 4299444 | 87689417 | 20.39 |
| Moesin | MSN | 12531147 | 33608958 | 2.68 |
| Histone H2, H3, H4 | H2,H3,H4 | 4117818 | 14248749 | 3.46 |
| Neutrophil elastase | ELANE | 5827421 | 13787790 | 2.36 |
| Peroxiredoxin-1 | PRDX1 | 2254814 | 7266159 | 3.22 |
| Annexin A2 | ANXA2 | 1486325 | 6293004 | 4.23 |
| Heat shock protein 60 | HSPD1 | 1872207 | 3309279 | 1.76 |
| S100 calcium-binding protein A9[#] | S100A9 | 0 | 1967366 | NA |
| Heat shock 70 kDa protein 1A | Hspa1a | 0 | 1637649 | NA |
| Heat shock protein 90 alpha/beta | HSP90aa1/b1/ab1 | 8455798 | 32192904 | 3.80 |
| Caveolin-1 | CAV1 | 45650 | 695619 | 15.23 |
| Histidine-tRNA ligase | HARS1 | 55924 | 294886 | 5.27 |
| Colfosceril Palmitate | PC 16:0/16:0 | 62107 | 103352 | 1.66 |

Values displayed as average abundance ($n = 3$). # = Biomolecules evaluated in subsequent testing.

Poly(ethylene glycol) methyl ether-block-poly(lactide-co-glycolide) (PLGA-PEG5k) (Product No. 900950), poly(vinyl alcohol) (Product No. P8136), LPS from Escherichia Coli serotype O111:B4, FITC-conjugated LPS from Escherichia coli serotype O111:B4, Bay 11-7082 (Product No. B5556), paquinimod (Product No. SML2883), human fibrinogen (Product No. F3879), bovine hemoglobin (Product No. H2500), RPMI 1640, and acetone were obtained from Sigma-Aldrich (St. Louis, MO). Tj-m2010-5 MyD88 inhibitor (Product No. HY-139397) was purchased from MedChemExpress (Monmouth Junction, NJ). Endotoxin quantification assay (Product No. A39552) and CBQCA assay (Product No. C6667) were purchased from Thermo Fisher Scientific (Waltham, MA). RAW-Blue™ cells and QUANTI-Blue™ solution (Cat. raw-sp) were purchased from InvivoGen (San Diego, CA). Female C57BL/6 mice (5–6 weeks old) were purchased from the University of Maryland Vet Resources (Baltimore, MD). TLR4[-/-] K/O C57BL/6 J (B6(Cg)-Tlr4[tm1.2Karp]/J) mice were gifted by S. Vogel Lab at the University of Maryland, Baltimore (Baltimore, MD). Mice were housed on a 12-hour light/12-hour dark cycle at ambient temperature (65–75ºC), and 40–60% relative humidity. LC-MS grade acetonitrile, methanol, water, and n-propanol were purchased from Fisher Scientific (Pittsburg, PA). HPLC grade tert-Butyl methyl ether (MTBE), chloroform, ammonium formate, and formic acid was purchased from Sigma Aldrich (St. Louis, MO). EquiSPLASH lipidomix was purchased from Avanti Polar Lipids, Inc. (Alabaster, AL).

### Microfluidics System
Using a microfluidics system (Dolomite, Royston, UK) equipped with a 5-input chip (150 μm channel) (Part No. 3200711), NPs were generated through hydrodynamic flow focusing nanoprecipitation with PLGA, PLA, PLGA-PEG5k (50:50), or PLGA-Cy5.5 as the core polymers dissolved in acetone at 10 mg/mL and E400 PEMA or PVA surfactants dissolved at 1 mg/mL in $H_2O$ as a stabilizer[37]. Flow rate ratio of 1 was used during NP synthesis to achieve consistent size distributions. PLGA and PLA NPs were then subjected to centrifugation at 13,000xg, while PLGA-PEG5k NPs were centrifuged at 50,000xg to eliminate excess surfactant by washing thrice with cold $H_2O$. Sucrose (4% w/v) and

mannitol (3% w/v) were added to the NP suspensions as a cryoprotectant. The NPs were then frozen at -80 °C and lyophilized for later storage at RT in a desiccator. The size and zeta potential of PLGA NPs and coronas were determined in 1x PBS by dynamic light scattering (DLS) using a Malvern Zetasizer Nano ZSP and Nanoparticle Tracking Analysis (NTA) using a Malvern Nanosight NS300 (Malvern, UK) set to $n = 3$ recording measurements at a flow rate of 20 μL/minute.

### Biomolecular Corona Formation
NPs suspended in 1x PBS at a concentration of 20 mg/mL were combined with 100% whole plasma at equal volumes, resulting in a final plasma concentration of 50%. Samples were allowed to incubate at 37 °C for 1 hour under slight agitation. After incubation, ice-cold 1x PBS was added to preserve NP coronas, and unbound material was separated through centrifugation at 13,000x $g$, 4 °C, for 30 min or 50,000x $g$ centrifugation in the case of PLGA-PEG5k NP coronas. Pelleted NP coronas were then washed three times through successive resuspension and centrifugation in ice-cold PBS at a volume 10-fold greater than the initial NP-corona solution. NP-corona pellets were washed thrice to ensure the removal of unbound impurities and a consistent hard corona sample[77]. Final NP corona pellets were resuspended in 1x PBS for use in subsequent experiments.

### Animal studies
LPS-induced endotoxemia mouse model: Female C57BL/6 J mice (6–8 weeks) ($n = 10$ per condition) were intraperitoneally injected with 20 mg/kg LPS[49,78]. Plasma was collected into K2 EDTA coated tubes and pooled at 3, 8, and 16-h post LPS injection for plasma analysis and subsequent ex-vivo NP corona formation. Mice were housed under specific pathogen-free conditions in a facility at the University of Maryland, Baltimore Veterinary Resources. All mice experiments were approved by the Institutional Animal Care and Use Committee (IACUC) of the University of Maryland, Baltimore (IACUC ID: AUP-00000975).

NP corona organ distribution: Cy5.5-labeled PLGA NP coronas were injected intravenously through the tail vein at 2 mg per mouse. Mice were then sacrificed after 3 h; organs and plasmas were isolated

for Xenogen in vivo imaging system (IVIS) analysis of organ distribution and ELISA plasma cytokine quantification. Total radiant efficiency was calculated for all organs on a per-mouse basis and used to calculate the percent of NP amount in each organ.

## Protein and Lipopolysaccharide (LPS) Quantification

To determine the concentration of total plasma protein, a Pierce BCA Protein Assay Kit (Thermo Fisher Scientific, Waltham, MA) was used. Plasma samples were diluted 1000-fold before protein quantification following manufacturers protocol. The protein concentration was determined by comparing the absorbance at 562 nm to a standard curve prepared using bovine serum albumin (BSA) using a SpectraMax iD3 Microplate Reader (Molecular Devices, San Jose, CA).

CBQCA assay for corona-protein quantification was performed following a modification of the manufacturer protocol[79]. PLGA NPs were incubated in various plasmas to form coronas as described above. NP coronas were then dissolved in DMSO at a concentration of 20 mg/mL to solubilize the polymer and adsorbed proteins. 10 μL of dissolved NP corona samples were mixed with 10 μL of 5 mM ATTO-TAG CBQCA reagent, 5 μL of 20 mM KCN, and 125 μL of 0.1 M sodium borate buffer (pH 9.3), incubated for 1 hour at room temperature, and measured for fluorescence at an excitation of 465 nm and emission at 550 nm.

Quantification of endotoxin was accomplished by a 1:1000 dilution of whole plasma or NP corona samples following the manufacturer's protocol.

## SDS-PAGE Corona Fingerprinting

After NP-corona third washing step, 0.5 mgs of NPs per condition were mixed with Laemmli buffer (0.277 M Tris-HCl, 44.4% Glycerol, 4.4% LDS, 2-mercaptoethanol, 0.02% bromophenol blue) and heated to 95 °C for 5 min. Samples were then centrifuged at 13,000 xg to separate denatured proteins from solid NPs, and supernatants containing eluted corona proteins were run through a 4%-15% gradient SDS/PAGE gel. A molecular weight protein standard (Product No. 1610394) from Bio-Rad (Hercules, CA) was run for reference. After completion, gels were stained for total protein with Coomassie for 2 h and destained in a solution of 50:40:10 water:methanol:acetic acid, following manufacturer guidelines. Images were taken using a ThermoFisher iBright imaging system and processed on ImageJ (Version 1.54 h).

## Proteomic Analysis

Following previously established methods, whole plasmas and NP-corona samples were lysed in 5% sodium dodecyl sulfate (Sigma, L4509), 50 mM triethylammonium bicarbonate (1 M, pH 8.0) (Sigma, 7408)[80,81]. Proteins were subsequently extracted and digested using S-trap micro columns (ProtiFi, NY). These eluted peptides were then dried, reconstituted in 0.1% formic acid, and a BCA assay kit (Thermo Fisher Scientific, 23275) was used to quantify the total peptide concentration. A nanoACQUITY Ultra-Performance Liquid Chromatography analytical column (BEH130 C18, 1.7 μm, 75 μm x 200 mm; Waters Corporation, Milford, MA, USA) was used to separate all tryptic peptides over a 185-min linear acetonitrile gradient (3–40%) that contained 0.1% formic acid on a nanoACQUITY Ultra-Performance Liquid Chromatography system (Waters Corporation, Milford, MA USA), which was analyzed on a coupled Orbitrap Fusion Lumos Tribrid mass spectrometer (Thermo Scientific, San Jose, CA USA). A resolution of 240,000 was used to acquire full scans and precursors were selected for fragmentation by high-energy collisional dissociation (35% for a maximum 3-s cycle). Raw data was processed with Proteome Discoverer (PD, version 2.5.0.400, Thermo Fisher Scientific) using the Sequest HT search engine against a UniProt mouse reference proteome (release 2022.06, 17096 entries). Searches were configured with static modifications for carbamidomethyl on cysteines (+57.021 Da), dynamic modifications for oxidation of methionine residues

(+15.995 Da), precursor mass tolerance of 20 ppm, and a fragment mass tolerance of 0.5 Da. Trypsin was used as a digestion enzyme with a maximum of two missed cleavages. Minimum and maximum peptide lengths were set as 6 and 144, respectively. Minora feature detector, a tool embedded in the PD bioinformatics platform, was used for label-free quantification[82]. Protein identification was filtered to achieve a 1% false discovery rate (FDR) in peptide spectra match (PSM), peptide, and protein levels. This FDR was determined using the Percolator algorithm embedded in PD. The resulting protein abundances were analyzed using Perseus software (version 1.6.14.0). For robust statistical analysis, data was further filtered to include only proteins identified without any missing values in all the biological replicates. The quantitative protein data were $\log_2$ transformed and normalized using median centering. Differentially expressed proteins from various conditions were identified using ANOVA-based analysis or Two-tailed student's t-test (adjusted p-value < 0.05).

## Lipid Extraction and Analysis

A modified MTBE lipid extraction protocol was used to extract total lipid extracts from samples[83,84]. Briefly, a mixture of 400 μL cold methanol and 10 μL internal standard (EquiSPLASH lipidomix) was added to each sample, followed by orbital shaking at 4 °C for 15 min. 500 μL of cold MTBE was then added, and the samples were incubated at 4 °C for 1 hour with orbital shaking. 500 μL of cold water was gradually added, and the resulting extract was incubated at 4 °C for 15 min with orbital shaking. Samples were then centrifuged at 6010x $g$ for 8 min at 4 °C to allow for phase separation. The upper organic phase was carefully collected and kept on ice, while the lower aqueous phase was re-extracted with 200 μL of MTBE followed by 15 min of incubation at 4 °C with orbital shaking. After an additional centrifugation at 6010x $g$ for 8 min at 4 °C, the upper organic phase was removed and combined with the first organic extract. Organic extracts were then dried under a steady stream of nitrogen at 30 °C to remove solvents, and the recovered lipids were dissolved in 200 μL of chloroform:methanol (1:1, v/v) containing 200 μM of butylated hydroxytoluene. Prior to analysis, samples were diluted 4 fold with acetonitrile:isopropanol:water (1:2:1, v/v/v). The lower aqueous phase was used to determine the protein content via a BCA kit (bicinchoninic acid assay, Thermo Fisher Scientific, Rockford, USA).

Liquid chromatography coupled to high-resolution tandem mass spectrometry (LC-MS/MS) on an Agilent 1290 Infinity LC coupled to an Agilent 6560 Quadrupole Time-of-Flight (Q-TOF) mass spectrometer was used to analyze total lipid extracts. The separation was carried out using a C18 CSH (1.7 μm; 2.1 ×100 mm) column (Waters, Milford, MA). Mobile phase A was 10 mM ammonium formate with 0.1% formic acid in water/acetonitrile (40:60, v/v) and mobile phase B was 10 mM ammonium formate with 0.1% formic acid in acetonitrile/isopropanol (10:90, v/v). The gradient was ramped from 40% to 43% B in 1 min, ramped to 50 % in 0.1 min, ramped to 54% B in 4.9 min, ramped to 70% in 0.1 min, and ramped to 99% B in 2.9 min. For column equilibration, the gradient was returned to initial conditions in 0.5 min and held constant for 1.6 min at a flow rate of 0.4 mL/min. The column was heated to a temperature of 55 °C and the auto-sampler was kept at 5 °C during the analysis. A 2 μL injection volume was used for all samples analyses. Two workflows were used for mass spectrometry analysis. First, lipid identification of a pooled sample using an iterative MS/MS acquisition. Second, lipid semi-quantitation of all samples using high-resolution, accurate mass MS1 acquisition. The MS parameters for the iterative workflow were employed as described: extended dynamic range, 2 GHz; gas temperature, 200 °C; gas flow, 10 L/min; nebulizer, 50 psi; sheath gas temperature, 300 °C; sheath gas flow, 12 L/min; VCap, 3.5 kV (+), 3.0 kV (−); nozzle voltage, 250 V; reference mass m/z 121.0509, m/z 1221.9906 (+), m/z 119.0363, m/z 980.0164 (−); MS and MS/MS Range m/z 100–1700; acquisition rate, 3 spectra/s; isolation, narrow ( ~ 1.3 m/z); collision energy 20 eV (+), 25 eV (−); max

precursors/cycle, 3; precursor abundance-based scan speed, 25,000 counts/spectrum; ms/ms threshold, 5,000 counts and 0.001%; active exclusion enabled yes; purity, stringency 70 %, cut off 0%; isotope model, common organic molecules; static exclusion ranges, m/z 40 to 151 (+,−). The MS parameters for the MS1 workflow were the same for source and reference mass parameters and differed only for acquisition (selection of MS (same parameters) not Auto MS/MS).

The LC-MS data from the iterative MS/MS workflow was analyzed using Agilent's Lipid Annotator (v 1.0) with the feature finding and identification parameters at default settings. Positive and negative ion mode adducts included [M + H]+, [M+Na]+, [M + NH4]+, [M-H]-, and [M + CH3CO2]-, respectively. Corresponding results obtained from Lipid Annotator were saved as a PCDL file. The LC-MS data from the MS1 workflow were processed using Agilent's MassHunter Profinder (v 10.0). Batch-targeted feature extraction using default parameters and the PCDL file created from Lipid Annotator were used for feature extraction. Processed data generated from Profinder that included the peak areas and lipid identifications was subsequently exported into MetaboAnalyst 5.0 for multivariate analysis[85]. Univariate analysis performed using Prism 6 (GraphPad, La Jolla, CA).

## Metabolomic Analysis

High throughput targeted and quantitative metabolomic analysis was carried out using the Biocrates MxP Quant 500 kit (Biocrates, Life Science AG, Innsbruck, Austria). The assay quantifies up to 630 metabolites and lipids by combining liquid chromatography (LC) tandem mass spectrometry (MS) and flow injection analysis (FIA) workflows. The assay covers a number of classes (number of molecules in each class): alkaloids (1), amine oxides (1), amino acids (20), amino acid related metabolites (30), bile acids (14), biogenic amines (9), carbohydrate and related metabolites (1), carboxylic acids (7), cresols (1), fatty acids (12), hormone and related metabolites (4), indoles and derivatives (4), nucleobases and related metabolites (2), vitamins and cofactors (1), acylcarnitines (40), lysophosphatidylcholines (14), phosphatidylcholines (76), sphingomyelins (15), ceramides (28), dihydroceramides (8), hexosylceramides (19), dihexosylceramides (19), trihexosylceramides (6), cholesterol esters (22), diglycerides (44), and triglycerides (242). The MxP Quant 500 kit was prepared following the instructions of the manufacturer. Plasma samples of 10 μl were loaded on the filters of 96 well kit plates and subjected to drying under the stream of nitrogen. Drying is followed by derivatization of biogenic amines and amino acids using a 5% phenylisothiocyanate (PITC) in ethanol: water: pyridine (1:1:1;v:v:v). Metabolite extraction was performed using five mM ammonium acetate in methanol. Metabolite detection is done by multiple reaction monitoring (MRM) transitions. The method employs either direct infusion coupled with tandem mass spectrometry (DI – MS/MS) or reverse phase liquid chromatography coupled with tandem mass spectrometry (LC – MS/MS) workflows using a Waters ACQUITY UPLC system (Milford, MA) coupled to a Waters TQ-XS tandem quadrupole mass spectrometer (Waters Corp., Milford, MA). The LC-MS/MS separation was achieved using a C18 CSH (1.7 μm; 2.1 ×100 mm) column (Waters, Milford, MA). Mobile phase A was 0.1% formic acid in water, and mobile phase B was 0.1% formic acid in acetonitrile. The gradient was ramped from 40% to 43% B in 2 min, 50% in 0.1 min, 54% B in 9.9 min, 70% in 0.1 min, and 99% B in 5.9 min. The gradient was returned to initial conditions in 0.5 min and held for 1.9 min for column equilibration. The flow rate was 0.5 mL/min. The column was maintained at 30 °C, and the auto-sampler was kept at 10 °C. A 2 μL injection was used for all samples. The Web IDQ software (Biocrates, Life Science AG, Innsbruck, Austria) was used to register samples, calibrate, calculate metabolite concentrations, and validate the assay. All reagents used were analytical grade, and the mobile phases used were LC-MS grade. The data generated from the Biocrates MxP Quant 500 kit was analyzed with MetIDQ software

(Biocrates Life Sciences AG, Innsbruck, Austria), which yielded the analyte name and calculated concentration in μM, which was then imported into MetaboAnalyst 5.0 for multivariate analysis. The metabolite data were sum normalized, log-transformed, and mean-centered. Statistical analyses included multivariate analysis performed using the MetaboAnalyst 5.0 web-based statistical package[85] univariate analysis using GraphPad Prism (v. 6.03; La Jolla, California, US).

## BMDM Differentiation

Bone marrow was isolated from the tibias and femurs of 5–7 week-old female C57BL/6 mice or C57BL/6 J TLR4 K/O mice for the differentiation of bone marrow-derived macrophages (BMDMs)[49]. Isolated bone marrow was then subjected to ACK lysis, and residual stem cells were cultured in complete RPMI 1640 media containing penicillin (100 units/mL), streptomycin (100 μg/mL), 10% heat-inactivated fetal bovine serum (VWR, Radnor, PA), and 20% L929 cell conditioned medium (containing M-CSF) to induce BMDM differentiation. BMDMs were allowed to differentiate for 8 days, with cell conditioned media changes on days 3 and 6.

## Cytokine Detection

BMDMs from Naïve mice or TLR4 K/O mice (1 × 10^5 cells/well) were seeded in a sterile 24-well plate and treated with 300 μg/mL of PLGA NPs or the different NP coronas at 37 °C and 5% CO2. Supernatants were taken after 3 h of incubation, unless stated otherwise. Cytokine profiles in cell culture supernatants and diluted mouse plasmas were analyzed using an enzyme-linked immunosorbent assay (ELISA) (BioLegend, San Diego, CA) for TNFα or a custom 7-plex Luminex panel for IL-1β, MCP-1, IL-6, TNFα, IL-10, CXCL1, and IFN-β following the manufacturer's protocols and analyzed using the Luminex xPONENT software (Thermo Fisher, Waltham, MA). Heatmap and PCA plots were generated using MetaboAnalyst 5.0, with mean data centering.

BAY 11-7082 was used to inhibit NF-κB where cells were pre-treated at 10 μM for 30 min prior to NP corona treatment for 3 h. TJ-2010-5 was used to inhibit MyD88 activation where cells were pre-treated at 30 μM for 2 h prior to NP corona treatment.

## NF-κB Reporter Assay

RAW-Blue cells were cultured in DMEM (4.5 g/L glucose, 2 mM L-glutamine, 10% FBS, 100 μg/ml Normocin™, Pen-Strep (100 U/mL) following the manufacturer protocol. Cells were treated with 300 μg/mL PLGA NPs or NP coronas. Supernatants from cells were collected 24-h post-stimulation. 20 μL of supernatant was mixed with 180 μL of QUANTI-Blue solution and allowed to react in the dark at 37 °C for two hours. Secreted alkaline phosphatase (SEAP) activity was measured by reading the absorbance at 620 nm using a microplate reader.

## Flow Cytometry Phenotype Analysis

Day 8 BMDMs (1 × 10^5 cells/well) were seeded in a sterile 24-well plate and treated with 300 μg/mL of Cyanine-5.5 labeled PLGA NP samples, or 10 μg/mL for uptake studies. After NP corona treatment for 30 min or 24 h, cells were washed with fresh media and lifted with Versene solution. Lifted cells were incubated with anti-CD16/32 (1:500, Cat No. 101302, Clone 93) antibody to block nonspecific binding before staining with anti-mouse CD11b-Pacific Blue (1:500, Cat No. 101224, Clone M1/70), F4/80-PE/Cyanine7 (1:500, Cat No. 123114, Clone BM8), CD86-APC/Cyanine7 (1:666.66, Cat No. 105030, Clone GL-1), CD80-FITC (1:100, Cat No. 104706, Clone 16-10A1), PD-L1-PE (1:200, Cat No. 124308, Clone 10 F.9G2) antibodies from BioLegend (San Diego, CA). Cell viability was measured with propidium Iodide (PI) from BioLegend. The stained cells were measured on a Cytek Aurora flow cytometer (Fremont, CA) and analyzed using FCS Express 7 De Novo Software (Glendale, CA).

## Ingenuity Pathway Analysis

P-values were corrected using a two-way ANOVA with a Benjamini-Hochberg algorithm FDR correction. The list of differentially abundant biomolecules (proteins, lipids, and metabolites) between 3hrPlas_PLGA and NaivePlas_PLGA coronas, alongside the cytokine Luminex data, were uploaded into the IPA software (Qiagen). The "core analysis" function was used to interpret the datasets. Causal network analysis was supervised based on the observed BMDM cytokine profiles in order to predict 3hrPlas corona-mediated pathway activation. Further TLR4 ligand prediction was done by using the pathway "grow" feature, restricted to our corona dataset, to identify direct binders from the 3hrPlas_PLGA NP coronas.

## Biomolecule Validation

Purified hemoglobin at 10 mg/mL and fibrinogen (Fb) at 2 mg/mL ($Fb_{Norm}$) or 10 mg/mL ($Fb_{High}$), were incubated with PLGA NPs at 37 °C for 1 hour. Additionally, purified hemoglobin or fibrinogen was spiked into Naïve plasma at a final concentration of 10 mg/mL, then incubated with NPs to attempt to simulate 3hrPlas concentrations. All NPs were then washed thrice with cold 1x PBS as described earlier and BMDM cells were treated with 300 μg/mL for 3 h before supernatant TNFα cytokine analysis.

Paquinimod was added into whole 3hr-post LPS plasma at a final concentration of 30 μM. Samples were allowed to incubate for 1 hour to allow for inhibitor binding. Inhibited 3 hr plasma was then used to coat PLGA NPs as stated above. Washed NP coronas were then used to treat BMDM cells at 300 μg/mL for 3 h before supernatant cytokine analysis.

## Statistics/Bioinformatics

The data analysis was performed using GraphPad Prism 10.4.1 (GraphPad Software Ins.). Results are presented as mean ± standard deviation of $n = 3$ independent experiments unless stated otherwise.

## Reporting summary

Further information on research design is available in the Nature Portfolio Reporting Summary linked to this article.

## Data availability

The proteomics data generated in this study have been deposited in the PRIDE database under accession code PXD050918. The lipidomics data generated in this study have been deposited in the Mendeley database [https://doi.org/10.17632/2z97z3bcw6.1]. The metabolomics data generated in this study have been deposited in the Mendeley database [https://doi.org/10.17632/4mfr9btjtt.1]. Source data is available for Figs. 1b, c, e, f, g, 2a, b, c, e, f, 3a, b, c, g, h, i, 4a, b, c, d, 5c, and Supplementary Figs. 1b, 2a, 2b, 2e, 4, 5, 7a, 7b, 8a, 8b, 8c, 9a, 9b. in the associate source data file. Source data are provided as a Source Data file. Source data are provided with this paper.

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

## Acknowledgements

Research reported in this publication was supported the National Institute of General Medical Sciences of the National Institutes of Health under award number R35GM142752 (R.M.P). PhRMA Foundation Predoctoral Fellowship in Drug Delivery under award number 2022 PDDL 877305 (N.T.). University of Maryland School of Medicine's & Greenebaum Comprehensive Cancer Center's Flow Cytometry Core–Baltimore, Maryland. This publication was supported by funds through the Maryland Department of Health's Cigarette Restitution Fund Program and the National Cancer Institute - Cancer Center Support Grant (CCSG) - P30CA134274. University of Maryland School of Medicine's Center for Translational Research in Imaging – Baltimore, Maryland. University of Maryland School of Pharmacy Mass Spectrometry Center (SOP1841-IQB2014). BioRender was used for preparing graphics. The content is solely the responsibility of the authors and does not necessarily represent the official view of the National Institutes of Health.

## Author contributions

J.R.S. and R.M.P. conceived the project and wrote the manuscript. J.R.S., N.C., and N.T. designed and performed the in vitro experiments. J.R.S. and N.C. performed in vivo experiments and sample processing. M.W., A.T., N.P., and S.P. performed the mass spectrometry studies with the supervision of M.A.K. and J.W.J. J.R.S. performed computational and statistical analyses. R.M.P acquired funding and supervised the project. All authors edited and approved the final manuscript.

## Competing interests

The authors declare no competing interests.
