## [Peer Review file · Nature Communications]

Inflammatory Disease Progression Shapes Nanoparticle Biomolecular Corona-Mediated Immune Activation Profiles

Corresponding Author: Professor Ryan Pearson

Version 0:

Reviewer comments:

Reviewer #1

(Remarks to the Author)

This is an interesting, informative, well written, high-quality study on the changes of PLGA nanoparticle coronas exposed to the blood of mice undergoing endotoxin shock. The molecular composition of the corona and mouse plasma were followed by a broad spectrum of analytical methods, including proteome, lipidome, and metabolomic measurements. The data showed major, disease-state-dependent proinflammatory effect of the corona, driven by TLR4/MyD88/NF- κ B signaling. A main message is the "illumination" of the personalized nature of corona formation under dynamic inflammatory conditions and the "NP-mediated immune activation profiles".

It would take long to scrutinize the dozens of presented changes of different proteins, lipids, and other molecules, among which the enrichment of the corona with low abundance plasma proteins, including TLR4 ligands, was stressed. The shortcoming of the ms that I see is exactly what the authors acknowledge, the uniqueness of the findings. Except for the global question, if the progress of diseases has an impact the corona of NPs, a self-evident answer is yes, without conducting any experiment.

As to what happens in 6–8-month-old C57BL/6 mice treated with 20 mg/kg LPS, which is near the LD-50; i.e., endotoxin shock, the animals are very sick. I am not aware of that PLGA, or any other nanomedicines, are intended to treat endotoxin shock. The study does not give information, how the observations relate to any disease that is treated with PLGA NPs, and it is not excluded that non-lethal, non-immune diseases do not influence the corona et al. That would mean the findings are clinically irrelevant.

At this stage the study is descriptive, and the title is misleading: "disease progression" implies an everyday disease, which cannot be equated with a life-threatening toxemia in severe infection with gram negative bacteria. Accordingly, before accepting this otherwise excellent study for publication, I recommend addressing the above concerns, either experimentally or theoretically. In the latter case, the authors should bring the conclusions closer to real life and address the universality, clinical relevance of their findings.

Reviewer #3

(Remarks to the Author)

The authors have addressed my concerns and suggestions. One can still find that the novelty is not at the highest level, but I would think that a publication in Nature Communications is a fair place for this publication.

Version 1:

Reviewer comments:

Reviewer #1

(Remarks to the Author)

Reviewer #1:

This is an interesting, informative, well written, high-quality study on the changes of PLGA nanoparticle coronas exposed to the blood of mice undergoing endotoxin shock. The molecular composition of the corona and mouse plasma were followed by a broad spectrum of analytical methods, including proteome, lipidome, and metabolomic measurements. The data showed major, disease-state-dependent proinflammatory effect of the corona, driven by TLR4/MyD88/NF- κ B signaling. A main message is the “illumination” of the personalized nature of corona formation under dynamic inflammatory conditions” and the “NP-mediated immune activation profiles”.

It would take long to scrutinize the dozens of presented changes of different proteins, lipids, and other molecules, among which the enrichment of the corona with low abundance plasma proteins, including TLR4 ligands, was stressed. The shortcoming of the ms that I see is exactly what the authors acknowledge, the uniqueness of the findings. Except for the global question, if the progress of diseases has an impact the corona of NPs, a self-evident answer is yes, without conducting any experiment.

As to what happens in 6–8-month-old C57BL/6 mice treated with 20 mg/kg LPS, which is near the LD-50; i.e., endotoxin shock, the animals are very sick. I am not aware of that PLGA, or any other nanomedicines, are intended to treat endotoxin shock. The study does not give information, how the observations relate to any disease that is treated with PLGA NPs, and it is not excluded that non-lethal, non-immune diseases do not influence the corona at all. That would mean the findings are clinically irrelevant.

We appreciate the reviewer’s thoughtful feedback and positive assessment of our study. We understand the reviewer’s point regarding the use of endotoxemia as a model inflammatory disease. We selected endotoxemia as a model pro-inflammatory disease, which allowed us to observe distinct immune signatures and enrichment patterns in the corona that may be more subtle under milder conditions but could provide mechanistic insights into NP behavior in inflammatory states.

While our lab, and others, have utilized PLGA/PLA NPs to treat endotoxin shock in preclinical models^{1,2} (and biomaterial-based approaches summarized³), we believe that our findings can provide insights relevant to other inflammatory disease states where immune activation and plasma protein alterations are present, such as in cancer, autoimmune diseases, chronic infections, and others. Moreover, the concept of the ‘personalized protein corona’ supports this notion as distinct protein fingerprints have been observed due to disease states⁴ as well as in different patients with the same disease⁵. Although disease progression is expected to change corona compositions, the distinct immune responses associated with these corona differences is often underappreciated and has not been previously evaluated. In recent years, there has been growing concern regarding the rigor and reproducibility of nanoparticle formulations, high interpatient variability in clinical trials, and their overall lack of successful clinical translation⁶. Thus, studies like ours can provide evidence that disease state/progression effects should be considered during the pre-clinical and clinical evaluation of nanomedicines. We have refined our manuscript to incorporate these points as described by the following changes:

Page 3: “Characterization of clinical formulations suggest that NPs concentrate disease-related acute inflammatory response proteins when introduced into blood from patients with systemic inflammatory conditions, like sepsis, which is not observed in healthy controls.^{19,20} Additionally,

ex vivo protein corona compositions demonstrate high interpatient variability and immune cell uptake within the same disease groups or otherwise healthy controls.^{11,21}

Page 4: “We employed lipopolysaccharide (LPS)-induced endotoxemia as a highly dynamic inflammatory disease state and model of severe systemic inflammation to evaluate NP corona formation and corresponding immunological effects altered by the acquired biological identity, which allowed us to observe distinct immune signatures and enrichment patterns in the corona that may be more subtle under milder conditions but could provide mechanistic insights into NP behavior in inflammatory states.”

Page 7: “PLGA/PLA NPs have been extensively used as drug delivery vehicles for the treatment of inflammatory diseases, including pre-clinical models of endotoxemia, allergy, experimental autoimmune encephalomyelitis, and several others with varying levels of success.^{48–50} Our group and others have shown that similar NPs possess inherent anti-inflammatory and immunomodulatory properties; however, the timing of NP administration was found to contribute to their ability to mitigate lethality in endotoxemia.^{51–53} This observation could, in part, be related to the formation of an immunomodulatory corona, which could impede their ability to reduce inflammation by altering cellular uptake and immune activation. Future studies may benefit from evaluating the influence of NP formulation and purification parameters, PEG-alternative coatings, and other strategies to control the potential formation of immunostimulatory NP coronas and improve NP efficacy.”

Page 14: “Second, although there is previous literature demonstrating disease-specific protein corona composition differences in human plasmas, evaluating the effects of disease dynamics on corona-dependent immune recognition in additional disease models could help fine-tune the administration of NP therapeutics by identifying an optimal therapeutic window where coronas do not induce immune activation or possibly limit therapeutic efficacy.”

At this stage the study is descriptive, and the title is misleading: "disease progression" implies an everyday disease, which cannot be equated with a life-threatening toxemia in severe infection with gram negative bacteria. Accordingly, before accepting this otherwise excellent study for publication, I recommend addressing the above concerns, either experimentally or theoretically. In the latter case, the authors should bring the conclusions closer to real life and address the universality, clinical relevance of their findings.

We thank the reviewer for their comment. To better reflect the study's focus, we have revised the title to emphasize that this study investigated the influence of inflammation progression on NP corona formation and its corresponding influence on immune activation profiles: “Inflammatory Disease Progression Shapes Nanoparticle Biomolecular Corona-Mediated Immune Activation Profiles”

Reviewer #3 (Remarks to the Author):

The authors have addressed my concerns and suggestions. One can still find that the novelty is not at the highest level, but I would think that a publication in Nature Communications is a fair place for this publication.

We are grateful to the reviewer and thank them for their support.

References

1. Casey LM, Kakade S, Decker JT, Rose JA, Deans K, Shea LD, Pearson RM. Cargo-less nanoparticles program innate immune cell responses to toll-like receptor activation. *Biomaterials*. 2019;218:119333. Epub 2019/07/14. doi: 10.1016/j.biomaterials.2019.119333. PubMed PMID: 31301576; PMCID: PMC6679939.
2. Truong N, Cottingham AL, Dharmaraj S, Shaw JR, Lasola JJM, Goodis CC, Fletcher S, Pearson RM. Multimodal nanoparticle-containing modified suberoylanilide hydroxamic acid polymer conjugates to mitigate immune dysfunction in severe inflammation. *Bioengineering & Translational Medicine*. 2023. doi: 10.1002/btm2.10611.
3. Lasola JJM, Kamdem H, McDaniel MW, Pearson RM. Biomaterial-Driven Immunomodulation: Cell Biology-Based Strategies to Mitigate Severe Inflammation and Sepsis. *Frontiers in Immunology*. 2020;11(1726). doi: 10.3389/fimmu.2020.01726; PMCID: PMC7418829.
4. Hajipour MJ, Laurent S, Aghaie A, Rezaee F, Mahmoudi M. Personalized protein coronas: a “key” factor at the nanobiointerface. *Biomaterials Science*. 2014;2(9):1210-21. doi: 10.1039/C4BM00131A.
5. Papafilippou L, Claxton A, Dark P, Kostarelos K, Hadjidemetriou M. Protein corona fingerprinting to differentiate sepsis from non-infectious systemic inflammation. *Nanoscale*. 2020;12(18):10240-53. doi: 10.1039/D0NR02788J.
6. Leong HS, Butler KS, Brinker CJ, Azzawi M, Conlan S, Dufes C, Owen A, Rannard S, Scott C, Chen C, Dobrovolskaia MA, Kozlov SV, Prina-Mello A, Schmid R, Wick P, Caputo F, Boisseau P, Crist RM, McNeil SE, Fadeel B, Tran L, Hansen SF, Hartmann NB, Clausen LPW, Skjolding LM, Baun A, Agerstrand M, Gu Z, Lamprou DA, Hoskins C, Huang L, Song W, Cao H, Liu X, Jandt KD, Jiang W, Kim BYS, Wheeler KE, Chetwynd AJ, Lynch I, Moghimi SM, Nel A, Xia T, Weiss PS, Sarmiento B, das Neves J, Santos HA, Santos L, Mitragotri S, Little S, Peer D, Amiji MM, Alonso MJ, Petri-Fink A, Balog S, Lee A, Drasler B, Rothen-Rutishauser B, Wilhelm S, Acar H, Harrison RG, Mao C, Mukherjee P, Ramesh R, McNally LR, Busatto S, Wolfram J, Bergese P, Ferrari M, Fang RH, Zhang L, Zheng J, Peng C, Du B, Yu M, Charron DM, Zheng G, Pastore C. On the issue of transparency and reproducibility in nanomedicine. *Nat Nanotechnol*. 2019;14(7):629-35. doi: 10.1038/s41565-019-0496-9. PubMed PMID: 31270452; PMCID: PMC6939883.